# TimeVAE: A Variational Auto-Encoder for Multivariate Time Series Generation

## Abstract

Recent work in synthetic data generation in the time-series domain has focused on the use of Generative Adversarial Networks. We propose a novel architecture for synthetically generating time-series data with the use of Variational Auto-Encoders (VAEs). The proposed architecture has several distinct properties: interpretability, ability to encode domain knowledge, and reduced training times. We evaluate data generation quality by similarity and predictability against four multivariate datasets. We experiment with varying sizes of training data to measure the impact of data availability on generation quality for our VAE method as well as several state-of-the-art data generation methods. Our results on similarity tests show that the VAE approach is able to accurately represent the temporal attributes of the original data. On next-step prediction tasks using generated data, the proposed VAE architecture consistently meets or exceeds performance of state-of-the-art data generation methods. While noise reduction may cause the generated data to deviate from original data, we demonstrate the resulting de-noised data can significantly improve performance for next-step prediction using generated data. Finally, the proposed architecture can incorporate domain-specific time-patterns such as polynomial trends and seasonalities to provide interpretable outputs. Such interpretability can be highly advantageous in applications requiring transparency of model outputs or where users desire to inject prior knowledge of time-series patterns into the generative model.

## 1 Introduction

Interest in generative models has increased considerably in recent years as the usage of deep learning in industry and academic has exploded. Data generators are useful in scenarios that involve unavailability of any or sufficient amount of real data, restrictions on data usage due to privacy reasons, need to simulate situations not yet encountered in reality, simulating exceptional cases, or the need to create datasets for specific testing scenarios such as presence of outliers or changepoints. Data generators also can help alleviate the limitation of deep-learning models that are data-hungry.

Generative models fall under two broad categories, 1) trained generators that learn from real-world data, and 2) generators that use Monte Carlo sampling methods with user-defined distributions. Generative Adversarial Networks (GAN), Variational Auto-Encoders (VAE), and language models such as GPT (Radford et al., 2019) fall under the first category. Simple examples of the Monte Carlo generation include methods used to generate the classic donut-problem (Beiden et al., 2003), or X-OR problem data (Gomes et al., 2006). The major benefit of trained generators is that they can faithfully represent the patterns observed in the real-world data without requiring manual analysis, but their primary drawback is they can require large training data sets and long training times in order to learn to simulate the real data. Monte Carlo methods on the other hand, while being manual to set up, are simpler and more convenient to use. The generated data from them is interpretable and these methods also enable users to control the generated data, inject subject matter expertise, and simulate specific situations. The limitation is that generated data can be substantially different from real-world data. Therefore, a hybrid method that combines both approaches would be valuable for many downstream tasks such as prediction, forecasting, classification, and testing.

In the domain of time-series data, synthetic data generation is a challenging task due to the temporal patterns in the data. The generative process must capture both the distributions in features as well

as the temporal relationships. Deep learning methods are especially well-suited for modeling such complex relationships. However, in many real-world cases involving time-series data, the amount of available data can be limited in terms of number of samples, or in length of history. Examples include stock market predictions involving companies soon after initial public offering or predicting a retail organization's staffing needs in newly opened locations. Such situations require a data generation method that functions well despite low volume of data and also allows users to introduce specific constructs of time-series patterns that are known to exist in the specific use-case.

Recent work for synthetic data generation has largely focused on use of GANs (Yoon et al., 2019; Esteban et al., 2017), primarily by using recurrent neural networks for both generation and discrimination. However, due to the complexity introduced by temporal relationships, the standard approach of real versus synthetic binary discrimination is insufficient to capture temporal dependencies. As a result, special mechanisms are required within GAN networks to overcome this challenge. An example of such a special mechanism is the combination of supervised training traditionally used in autoregressive models with the unsupervised training of GAN in Yoon et al. (2019).

In order to leverage the strengths of both general data generation approaches we propose a new model based on Variational Auto-Encoders (VAEs) with a decoder design that enables user-defined distributions, which we hereafter refer to as TimeVAE. We demonstrate that TimeVAE can accurately model the temporal components of real-world data. Furthermore, we show that the bottleneck mechanism between the encoder and decoder that is inherent to VAEs acts to denoise the data which may help downstream tasks such as forecasting. Our method allows injection of custom temporal constructs such as level, trend, and seasonality to produce interpretable signals. Our experiments demonstrate TimeVAE meets the performance of top generative models when measuring similarity with original data, is computationally efficient to train, and outperforms available methods on the next-step prediction task. Finally, we show that our proposed method is superior to existing methods as the size of the available real-world training data diminishes.

## 2 RELATED WORKS

In Yoon et al. (2019), an unsupervised GAN approach is combined with the power of autoregressive models, creating TimeGAN. The GAN makes up for the deficiencies that autoregressive models have, namely that they are deterministic. GANs, however, struggle to adhere to the temporal correlations in time series data which autoregressive models excel at. This method is the current state-of-the-art in synthetic timeseries data generation.

Recurrent Conditional GAN (RCGAN) was invented to generate eICU data for privacy (Esteban et al., 2017). A recurrent GAN architecture is used for generating real-valued sequential data. More specifically, RNNs are used for both the generator and discriminator and are conditioned on auxiliary information, consisting of real-valued time-series data with associated labels.

C-RNN-GAN was developed to generate classical music data (Mogren, 2016). Randomly generated noise is input to deep LSTM modules to generate hidden representations which are then sent to the fully-connected layers to get the final generated sequential data samples. These samples and also the real music samples are then sent to the discriminator. This adversarial training helps the RNN generate music that varies in the number of tones used and the intensities.

Fabius & Van Amersfoort (2014) implemented a recurrent VAE which used RNNs for the encoder and decoder. The architecture used the last encoded state of the encoder RNN as representational of the input sequence feeding into the reparameterization trick used in VAEs. The sample from the encoded representation passes into the decoder. However, Hyland et all Esteban et al. (2017) reported inconsistent results when tested on generated sine wave data, with results being clearly inferior to those obtained by the RCGAN method.

Common limitations of the above generative models include: (1) difficult to train: the trained model is commonly subject to mode collapse, i.e. learning to sample synthetic data without diversity (Srivastava et al., 2017); (2) time consuming: the TimeGAN and RCGAN methods needed more than a day to be trained with 5,000 epochs by a V100 GPU on larger datasets; (3) Need sufficient data to train GAN-based models. It is not easy to obtain sufficient data in many real-world forecasting situations. Our proposed model attempts to address these limitations.

## 3 METHODS

Our goal is to meet the following two objectives. First, we want to select an architecture for the encoder and decoder components in the VAE that maximizes our ability to generate realistic samples of timeseries data. Second, we want the architecture that allows us to inject specific temporal structures to the data generation process. These temporal structures can be used to create an interpretable generative process and inject domain expertise in cases where we have insufficient real-world data to train the generator model.

### 3.1 VARIATIONAL AUTO-ENCODER AS A GENERATIVE MODEL

We will first provide a brief overview of VAEs. The encoder in a VAE outputs the distribution of the embedding instead of a point estimate as in a traditional Auto-Encoder. We are given an input dataset $X$ which consists of $N$ i.i.d. samples of some continuous or discrete variable $x$. Our main goal is to generate samples that accurately represent distributions in original data. Specifically, we want to model the unknown probability function $p(x)$.

The VAE described by Kingma & Welling (2013) assumes that the observed data are generated by a two step-process. First, a value $z$ is generated from some prior distribution $p_\theta(z)$, and second, a value $x$ is generated from some conditional distribution $p_\theta(x|z)$. While true values of the prior $p_\theta(z)$ and the likelihood $p_\theta(x|z)$ are unknown to us, we assume they are differentiable WRT both $\theta$ and $z$. Now, we can define the relationships between the input data and the latent representation as follows: the prior is $p_\theta(z)$, the likelihood is $p_\theta(x|z)$, and the posterior is $p_\theta(z|x)$.

The computation of $p_\theta(z)$ is very expensive and, quite commonly, intractable. To overcome this the VAE introduces an approximation of the posterior distribution as follows:

$$q_\Phi(z|x) \approx p_\theta(z|x)$$

With this framework, the encoder serves to model the probabilistic posterior distribution $q_\Phi(z|x)$, while the decoder serves to model the conditional likelihood $p_\theta(x|z)$.

Typically, this prior distribution of $z$ is chosen to be the Gaussian distribution, more specifically, the standard normal. Then the posterior is regularized during training to ensure that the latent space is wrapped around this prior. We do this by adding the KL divergence between the variational approximation of the posterior and the chosen prior to the loss function.

Since we embed the given inputs into a chosen prior, we can conveniently sample $z$ directly from the prior distribution and then pass this $z$ through the decoder. This essentially converts the VAE's decoder into a generative model.

### 3.2 VAE LOSS FUNCTION

The loss function for the VAE, also named Evidence Lower Bound loss function (ELBO), can be written as follows:

$$L_{\theta,\phi} = -E_{q_\phi(z|x)}[log p_\theta(x|z)] + D_{KL}(q_\phi(z|x)||p_\theta(z))$$

The first term on the RHS is the negative log-likelihood of our data given $z$ sampled from $q_\phi(z|x)$. The second term on the RHS is the KL-Divergence loss between the encoded latent space distribution and the prior. Note that the process involves sampling $z$ from $q_\phi(z|x)$ which would normally make the operation non-differentiable. However, VAEs use the so-called reparameterization trick which makes the VAE end-to-end trainable (Kingma & Welling, 2013).

### 3.3 BASE TIMEVAE ARCHITECTURE

Note that we have not yet described the encoding and decoding functions. One may choose any models for these as long as the loss function described above is differentiable. Our method uses a combination of traditional deep learning layers such as dense and convolutional layers and custom layers to model time-series specific components such as level, multi-polynomial trend, and seasonal patterns. Figure 1 provides a block-diagram of the base version of TimeVAE. The base version excludes the custom temporal structures. It does not require any time-series specific knowledge.

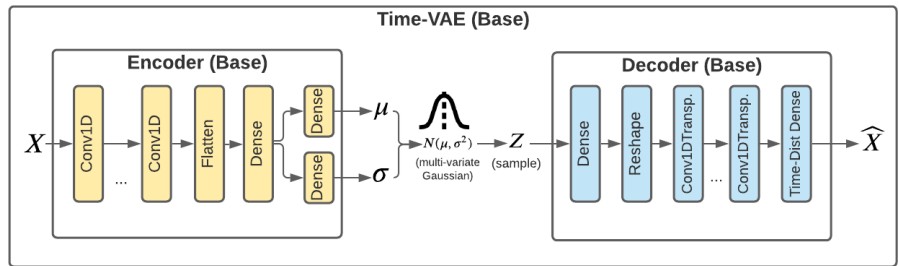

Figure 1: Block diagram of components in Base TimeVAE

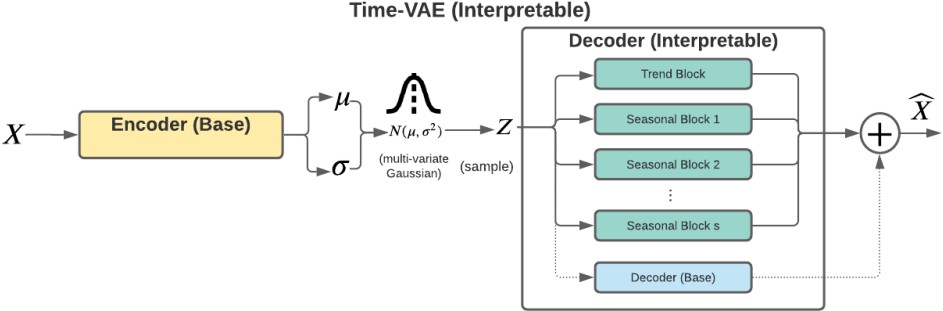

Figure 2: Block diagram of components in Interpretable TimeVAE

The input signal $X$ into the encoder is a 3-dimensional array of size $N \times T \times D$, where N is the batch size, T is the number of time steps, and D is number of feature dimensions. If given data have variable length sequences, then the sequences are padded with zeroes at the beginning to ensure all sequences have same length T. The encoder passes the inputs through a series of convolutional layers with ReLU activation. Next, the data are flattened before passing through a fully-connected (dense) linear layer. If $m$ is the number of chosen latent dimensions, representing dimensions of the multi-variate Gaussian, then this last layer has $2m$ number of neurons. We use the output to parameterize the multivariate Gaussian. The size of latent space $m$ is a key model hyper-parameter.

Next, we sample the vector $z$ from the multivariate Gaussian using the reparameterization trick. The decoder takes the sampled latent vector $z$ which is of length $m$. It is passed through a fully-connected linear layer. Then the data are reshaped into 3-dimensional array before passing through a series of transposed convolutional layers with ReLU activation. Finally, the data passes through a time-distributed fully connected layer with dimensions such that the final output shape is the same as the original signal $X$.

## 3.4 INTERPRETABLE TIMEVAE

We achieve interpretability of the modeled data generation process by injecting temporal structures to the data generation process in the decoder. More specifically, we use decomposition of time series into level, trend and seasonal components, a common approach in forecasting models; for example the well-known Holt-Winters exponential smoothing technique (Holt, 1957; Winters, 1960).

The Interpretable TimeVAE (Figure 2) uses the same encoder structure as that of the Base TimeVAE. The decoder has a more complex architecture because it imparts specific temporal structures to the decoding process. The decoder uses parallel blocks representing different temporal structures that are added together to produce the final output. The structures are of two types, namely, trend, and seasonality.

We next define the trend and seasonality blocks (see Figure 3). For sections below, let $N$ be the batch size, $D$ be the number of features, $T$ be the number of time steps (i.e. number of epochs).

**Trend Block:** We reuse the trend architecture developed for the N-Beats forecasting model by Oreshkin et al. (2019). Trend is modeled as a monotonic function. Let $P$ be the number of degrees of

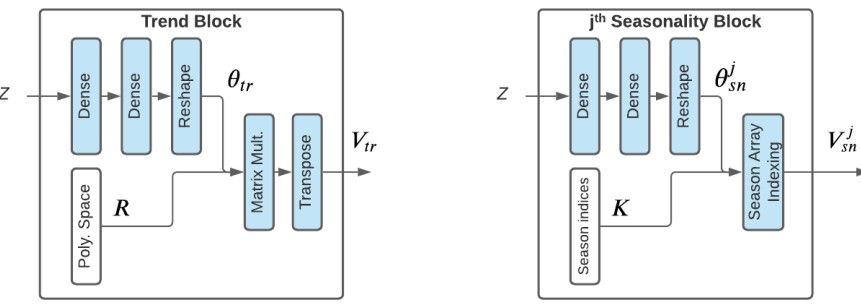

Figure 3: Trend and Seasonality Blocks in Interpretable TimeVAE

polynomial specified by the user. We model the trend polynomials $p = 0, 1, 2, .., P$ as a two step process. First, we use the latent space vector $z$ to estimate the basis expansion coefficients $\theta_{tr}$ for trend. $\theta_{tr}$ is of dimensionality $N \times D \times P$. Next, we use $\theta_{tr}$ to reconstruct the trend $V_{tr}$ in the original signal. The trend reconstruction in the signal in matrix form is as follows:

$$V_{tr} = \theta_{tr} R$$

$R = [1, r, ..., r^p]$ is the matrix of powers of $r$ where $r = [0, 1, 2, ..., T-1]/T$ is a time vector. $R$ is of dimensionality $P \times T$. We perform a matrix multiplication of $\theta_{tr}$ and $R$ and then transpose axes 1 and 2, resulting in final trend matrix $V_{tr}$ which has the dimensionality $N \times T \times D$.

The matrix $\theta_{tr}$ renders interpretability to the Trend block. Values from $\theta_{tr}$ specifically define the $0^{th}, 1^{st}, 2^{nd}, ...P^{th}$ order trend for each sample $N$ and feature dimension $D$.

Note that at $p = 0$, we get a flat trend (i.e. no upward/downward trend), which is equivalent to the level component in traditional time-series modeling nomenclature. Level refers to the average value of the series.

**Seasonality Block:** Let $S$ be the number of different seasonality patterns to be modeled. Each seasonality pattern, indexed by $j$, is defined by two parameters: $m$ as the number of seasons, and $d$ as the duration of each season. For example, to represent day-of-the-week seasonality for daily data, $m$ is 7 and $d$ is 1. On the other hand, for hourly level data, day-of-the-week seasonality is modeled with $m$ equal to 7 and $d$ equal to 24.

For each seasonality pattern $j$, TimeVAE performs two steps. First, the latent space vector $z$ is used to estimate the matrix of basis expansion coefficients $\theta_{sn}^j$ which has the dimensionality $N \times D \times m$. Next we index the elements in $\theta_{sn}^j$ corresponding to specific season for each time-step of $X$ to retrieve the seasonal pattern values $V_{sn}^j$ which is of shape $N \times T \times D$. In TensorFlow, we use the `gather` function to perform this indexing using the season indexing array $K$. The final seasonality estimates $V_{sn}$ are the element-wise summation of all $V_{sn}^j$ over $j = 1, 2, ..., S$.

In the seasonal block $j$, the matrix $\theta_{sn}^j$ provides interpretability of the seasonal pattern for each sample $N$ and feature dimension $D$. One can index elements in $\theta_{sn}^j$ to identify impact of each of the $m$ seasons within the seasonal cycle.

**Base Decoder as a Residual Block:** The Interpretable TimeVAE architecture also allows use of the original Base Decoder (Figure 1) as a residual branch in the decoder. In fact, one may choose to enable or disable any of the Trend, Seasonality, or Base Decoder branches shown in Figure 2.

**Interpretable Decoder Output:** The final output from the Interpretable Decoder is the element-wise summation of the Trend block output $V_{tr}$, seasonality block outputs $V_{sn}^j$ for $j = 1, 2, ..., S$, and the output from the residual Base Decoder (if used).

### 3.5 TimeVAE Objective Function

We train TimeVAE using the ELBO loss function defined earlier with one modification. We use a weight on the reconstruction error which works to increase or decrease the emphasis placed on the reconstruction loss compared to the KL-Divergence loss between approximated posterior $q_\phi(z|x)$ and the prior $p_\theta(z)$. Weight factor on reconstruction error ranged between 0.5 and 3.5 during our

following experiments and can be chosen by visually inspecting quality of generated samples, or through hyper-parameter tuning if the generated samples are used in a supervised learning task in a downstream application.

# 4 EXPERIMENTS

## 4.1 DATA SETS

We consider 4 multivariate data sets[1] in our experiments. The data sets include:

1. **sines:** We generate $10,000$ samples of a 5-dimensional sinusoidal sequence of varying frequencies, amplitudes, and phases where each feature is correlated with others. For each dimension $i \in \{1, 2.., 5\}$, $x_i(t) = asin(2\pi\eta t + \theta)$ where $\eta \sim u[0.1, 0.15]$, $\theta \sim u[0, 2\pi]$ and $a \sim u[1.0, 3.0]$.
2. **stockv:** $3,919$ samples (10 years) of 6-dimensional daily values of a stock from Yahoo Finance.
3. **energy:** $19,711$ samples of a 28-dimensional appliances energy prediction data set consisting of continuous-valued measurements from the UCI Machine Learning repository.
4. **air:** $9,333$ samples of 15-features of hourly-averaged responses from air quality sensors from the UCI Machine Learning repository.

## 4.2 COMPARISON METHODOLOGIES

We compare our VAE framework to RCGAN, a RNN trained with teacher-forcing (T-Forcing), and TimeGAN. These were the top performing methods compared to TimeGAN in Yoon et al. (2019)[2].

T-Forcing, or teacher-forcing, is a deterministic technique for training an autoregressive recurrent neural network where the ground truth is used as input instead of model output from a prior time step (Yoon et al., 2019). As Yoon et al. (2019) do not provide their version of T-Forcing code, we follow the supplementary description of their implementation to the best of our ability: 3-layer GRUs with hidden dimensions 4 times the size of the input features, tanh activation function, sigmoid as the output layer activation function, [0,1] min-max scaled outputs, $\lambda = 1$, and $\eta = 10$.

Instead of a deterministic autoregressive approach, RCGAN as mentioned in Esteban et al. (2017) applies a GAN architecture to sequential data but also drops dependence on the previous output while conditioning on additional input. We make use of the RCGAN code in `https://github.com/ratschlab/RGAN`, adjusting the settings to account for dimension and sample size of the different data sets.

TimeGAN is a generative time-series model combining adversarial GAN methodologies with autoregressive model techniques to account for temporal dynamics. We directly apply the authors' code on `https://github.com/jsyoon0823/TimeGAN`.

## 4.3 COMPARISON METRICS

To assess the quality of the generated data, we analyse 2-dimensional t-SNE plots of the original and synthetic data where the temporal dimension is flattened. We also train a classification model to distinguish between the original and synthetic data as a supervised task. The *discriminative score* is $(accuracy - 0.5)$ on the held-out set. A score close to 0 is better indicating the generated data is hard to distinguish from original data. Finally, we consider *predictive scores*; more specifically, we train a post-hoc sequence model to predict next-step temporal vectors. This 2-layer LSTM is trained on synthetic data and evaluated using the mean absolute error on the original data set.

These comparison metrics were all considered in Yoon et al. (2019), and we use the authors' available code for these metrics on `https://github.com/jsyoon0823/TimeGAN`. The lower the scores, the better.

---

[1]A link to data sets and experiment source codes will be made available here on publication.

[2]The source code for C-RNN-GAN was obtained from Mogren (2016), but we were unable to reproduce similar performance numbers reported in Yoon et al. (2019) and observed that the method was very unstable at the tasks. Therefore, we chose to exclude this method from our experiments.

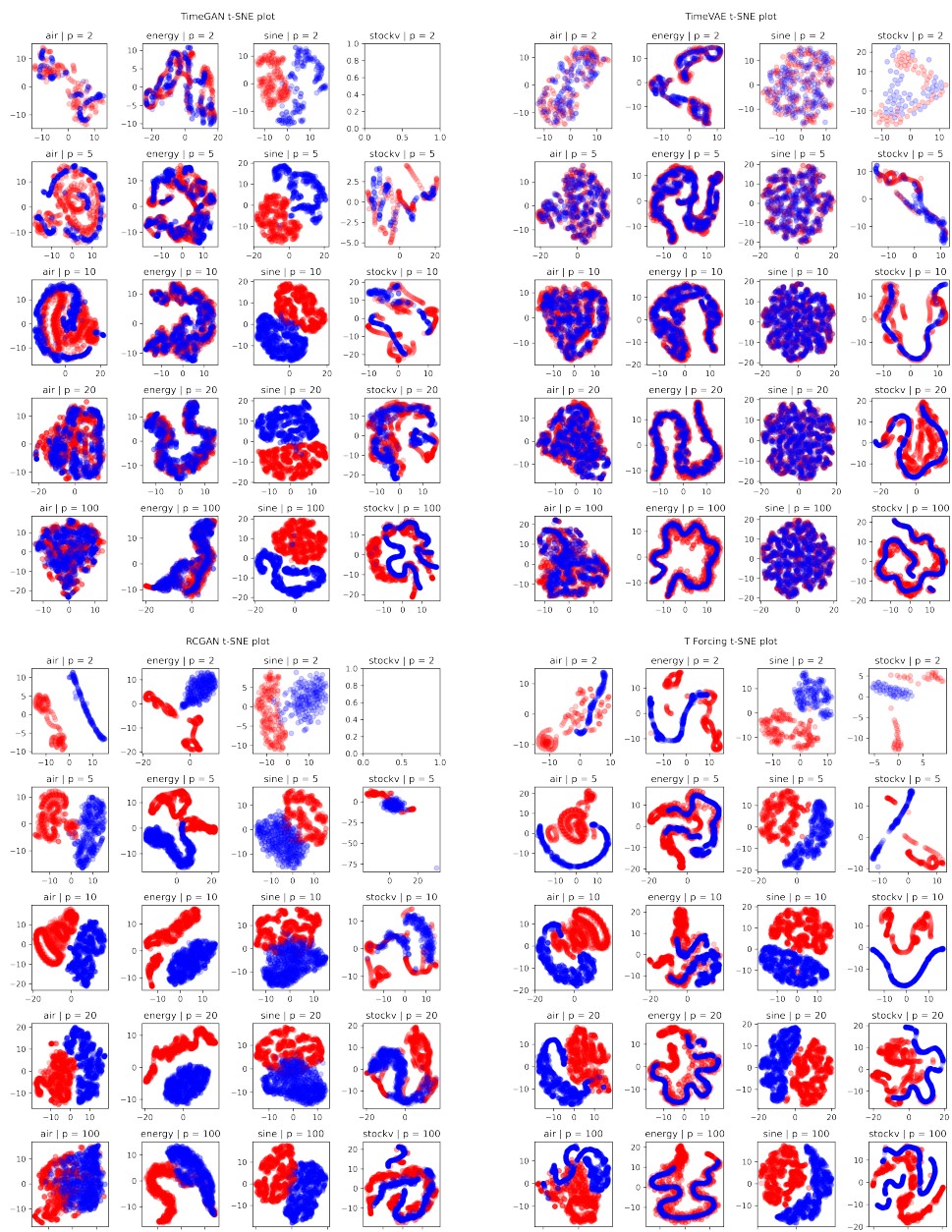

Figure 4: t-SNE plots for the TimeGAN (top left), TimeVAE (top right), RCGAN (bottom left), and T-Forcing (bottom right) models for the 4 data sets under various training size percentages ($p$). Red is for original data, and blue for synthetic data. An empty t-SNE chart appears when not enough training data is present for the model to generate synthetic data.

## 4.4 PROCEDURE

For each data set mentioned in subsection 4.1, we use 100%, 20%, 10%, 5%, and 2% as training data for each methodology. For example, we consider how well RCGAN performs when only trained on the last 20% of the air data set. The synthetic data generated by these trained models is then used to train the post-hoc sequence models to obtain discrimination and prediction scores described in subsection 4.3. The amount of data generated for training the post-hoc sequence models is equivalent to the percentage of original training data used to train the generators.

Table 1: Discriminator scores for all data set, model, and training percentages. N/A's exist when not enough data was available for the model to generate synthetic data. Best performance is in bold.

| model | % train | air | energy | sine | stockv |
|---|---|---|---|---|---|
| TimeVAE | 100 | **0.381 +/- 0.037** | **0.498 +/- 0.006** | **0.021 +/- 0.040** | **0.009 +/- 0.009** |
| TimeGAN | | 0.387 +/- 0.026 | 0.499 +/- 0.000 | 0.437 +/- 0.023 | 0.011 +/- 0.013 |
| T-Forcing | | 0.495 +/- 0.010 | 0.499 +/- 0.001 | 0.484 +/- 0.006 | 0.450 +/- 0.099 |
| RCGAN | | 0.495 +/- 0.002 | 0.500 +/- 0.000 | 0.382 +/- 0.075 | 0.494 +/- 0.006 |
| TimeVAE | 20 | **0.350 +/- 0.089** | 0.499 +/- 0.002 | **0.039 +/- 0.030** | 0.176 +/- 0.208 |
| TimeGAN | | 0.355 +/- 0.045 | **0.493 +/- 0.007** | 0.374 +/- 0.102 | **0.042 +/- 0.068** |
| T-Forcing | | 0.500 +/- 0.000 | 0.500 +/- 0.001 | 0.490 +/- 0.003 | 0.372 +/- 0.241 |
| RCGAN | | 0.500 +/- 0.000 | 0.500 +/- 0.000 | 0.281 +/- 0.132 | 0.479 +/- 0.028 |
| TimeVAE | 10 | 0.425 +/- 0.067 | **0.499 +/- 0.001** | **0.053 +/- 0.045** | 0.080 +/- 0.108 |
| TimeGAN | | **0.257 +/- 0.093** | 0.500 +/- 0.001 | 0.382 +/- 0.055 | **0.068 +/- 0.106** |
| T-Forcing | | 0.500 +/- 0.000 | 0.500 +/- 0.001 | 0.482 +/- 0.006 | 0.474 +/- 0.071 |
| RCGAN | | 0.500 +/- 0.000 | 0.500 +/- 0.000 | 0.246 +/- 0.234 | 0.474 +/- 0.071 |
| TimeVAE | 5 | **0.292 +/- 0.207** | 0.500 +/- 0.001 | **0.051 +/- 0.068** | 0.191 +/- 0.141 |
| TimeGAN | | 0.355 +/- 0.230 | **0.497 +/- 0.003** | 0.281 +/- 0.185 | **0.040 +/- 0.029** |
| T-Forcing | | 0.497 +/- 0.006 | 0.499 +/- 0.003 | 0.484 +/- 0.014 | 0.449 +/- 0.069 |
| RCGAN | | 0.500 +/- 0.000 | 0.500 +/- 0.000 | 0.499 +/- 0.003 | 0.500 +/- 0.000 |
| TimeVAE | 2 | **0.154 +/- 0.163** | 0.492 +/- 0.018 | **0.048 +/- 0.058** | **0.300 +/- 0.147** |
| TimeGAN | | 0.167 +/- 0.242 | **0.468 +/- 0.062** | 0.192 +/- 0.273 | N/A |
| T-Forcing | | 0.500 +/- 0.000 | 0.500 +/- 0.000 | 0.295 +/- 0.320 | 0.300 +/- 0.316 |
| RCGAN | | 0.494 +/- 0.017 | 0.500 +/- 0.000 | 0.492 +/- 0.021 | N/A |

## 5 RESULTS/DISCUSSION

Figure 4 displays the t-SNE charts of data generated from various generators for each of the datasets and training thresholds. The TimeVAE generated data consistently shows heavy overlap with original data for all datasets and training thresholds even at the 2% training threshold. We see de-noising in effect particularly for noisy datasets such as stockv and energy. Compared to T-Forcing and RC-GAN, TimeGAN has superior results on the air, energy and stockv datasets. However, performance of TimeGAN is sub-optimal on the sine dataset. In fact, all methods except TimeVAE had inferior-quality generated data on the sine dataset. One can also note that performance of other generators deteriorates at smaller training sizes. For example, while TimeGAN produces good quality generated data on the air dataset at the 100% threshold, the quality is inferior at thresholds of 2%, 5%, and 10%. For these reasons, the t-SNE plots indicate that the synthetic data generated by TimeVAE is superior compared to generated data from other methods.

Table 1 contains the discriminator scores for all scenarios. At the 100% threshold, TimeVAE produces the best results on the air, sine and stockv datasets. All generators perform poorly on the discrimination test on the energy dataset. Note that the energy dataset has 28 features and it is also the largest in number of samples. As was noted from the t-SNE charts, TimeVAE performs significantly better than other generators on the sine dataset. At lower thresholds of training sizes, results clearly show that TimeVAE and TimeGAN methods are superior to T-Forcing and RCGAN. These results do not conclusively identify the better method between TimeVAE and TimeGAN, except on the sine dataset. However, TimeGAN and RCGAN could not train successfully on 2% of stockv.

Results from the predictor tests are displayed in Table 2. The predictor scores measure Mean Absolute Error on the next-step prediction tasks. Results show that the TimeVAE method consistently meets or exceeds the performance from the other generators. On the sine dataset, TimeVAE produces the best results. Quite remarkably, performance of TimeVAE is almost on-par with those of the original datasets themselves on the stockv and sine datasets, even at the 2% threshold. In cases where TimeVAE does not have the best MAE score for other datasets, it is very close. For example, on the stockv dataset, TimeVAE and TimeGAN are indistinguishable in performance within the confidence interval. Similarly, on the energy dataset, TimeVAE and T-Forcing are the best methods.

Table 2: Predictor scores for all data set, model, and training percentages. *Original* means the LSTM for the predictor was trained using original data instead of synthetic data generated from any model and should always return the best performance possible. Best performance that is NOT *Original* is in bold. N/A's exist when not enough data was available for the model to generate synthetic data.

| model | % train | air | energy | sine | stockv |
|-------|---------|-----|--------|------|--------|
| Original | 100 | 0.004 +/- 0.000 | 0.229 +/- 0.002 | 0.213 +/- 0.000 | 0.019 +/- 0.004 |
| TimeVAE | | 0.013 +/- 0.002 | **0.268 +/- 0.004** | **0.213 +/- 0.000** | **0.019 +/- 0.001** |
| TimeGAN | | **0.005 +/- 0.000** | 0.298 +/- 0.002 | 0.251 +/- 0.027 | 0.021 +/- 0.001 |
| T-Forcing | | 0.101 +/- 0.028 | 0.287 +/- 0.035 | 0.220 +/- 0.010 | 0.082 +/- 0.004 |
| RCGAN | | 0.043 +/- 0.002 | 0.277 +/- 0.011 | 0.262 +/- 0.024 | 0.025 +/- 0.002 |
| Original | 20 | 0.004 +/- 0.001 | 0.187 +/- 0.003 | 0.215 +/- 0.000 | 0.049 +/- 0.001 |
| TimeVAE | | 0.019 +/- 0.003 | 0.288 +/- 0.002 | **0.215 +/- 0.000** | 0.052 +/- 0.001 |
| TimeGAN | | **0.007 +/- 0.002** | 0.324 +/- 0.005 | 0.287 +/- 0.051 | **0.050 +/- 0.001** |
| T-Forcing | | 0.139 +/- 0.061 | **0.256 +/- 0.006** | 0.219 +/- 0.007 | 0.091 +/- 0.024 |
| RCGAN | | 0.480 +/- 0.315 | 0.751 +/- 0.434 | 0.254 +/- 0.001 | 0.164 +/- 0.122 |
| Original | 10 | 0.002 +/- 0.001 | 0.162 +/- 0.004 | 0.214 +/- 0.000 | 0.072 +/- 0.000 |
| TimeVAE | | 0.005 +/- 0.003 | 0.275 +/- 0.001 | **0.215 +/- 0.000** | **0.075 +/- 0.001** |
| TimeGAN | | **0.003 +/- 0.001** | 0.318 +/- 0.006 | 0.300 +/- 0.059 | 0.081 +/- 0.008 |
| T-Forcing | | 0.157 +/- 0.027 | **0.262 +/- 0.014** | 0.216 +/- 0.002 | 0.118 +/- 0.040 |
| RCGAN | | 0.605 +/- 0.618 | 0.740 +/- 0.371 | 0.241 +/- 0.030 | 0.150 +/- 0.094 |
| Original | 5 | 0.022 +/- 0.001 | 0.144 +/- 0.015 | 0.217 +/- 0.000 | 0.078 +/- 0.002 |
| TimeVAE | | **0.040 +/- 0.002** | **0.262 +/- 0.002** | **0.218 +/- 0.001** | 0.084 +/- 0.004 |
| TimeGAN | | 0.046 +/- 0.005 | 0.329 +/- 0.010 | 0.262 +/- 0.032 | **0.080 +/- 0.001** |
| T-Forcing | | 0.185 +/- 0.012 | 0.263 +/- 0.011 | 0.221 +/- 0.006 | 0.131 +/- 0.026 |
| RCGAN | | 0.428 +/- 0.045 | 0.499 +/- 0.054 | 0.222 +/- 0.002 | 0.669 +/- 0.252 |
| Original | 2 | 0.026 +/- 0.003 | 0.106 +/- 0.002 | 0.222 +/- 0.002 | 0.144 +/- 0.006 |
| TimeVAE | | **0.056 +/- 0.005** | **0.260 +/- 0.003** | **0.223 +/- 0.001** | **0.153 +/- 0.008** |
| TimeGAN | | 0.057 +/- 0.003 | 0.313 +/- 0.003 | 0.270 +/- 0.004 | N/A |
| T-Forcing | | 0.185 +/- 0.086 | 0.266 +/- 0.005 | 0.225 +/- 0.005 | 0.155 +/- 0.003 |
| RCGAN | | 0.361 +/- 0.079 | 1.451 +/- 0.024 | 0.320 +/- 0.005 | N/A |

Table 3: Training times (in seconds) for all models using 100% of the data. Times were all obtained using the same ml.g4dn.xlarge AWS instance with 4 vCPU, 1 V100 GPU, and 16 GB of memory.

| model | air | energy | sine | stockv |
|-------|-----|--------|------|--------|
| TimeVAE | 442.39 | 1,661.40 | 386.20 | 95.40 |
| TimeGAN | 3,042.88 | 3,174.45 | 3,618.66 | 3,035.34 |
| T-Forcing | 115.14 | 134.91 | 117.59 | 109.19 |
| RCGAN | 9,102.56 | 23,806.19 | 4,365.80 | 1,442.70 |

## 6 CONCLUSIONS

In this paper, we introduce a novel Variational Auto-Encoder architecture for generating multivariate time-series data. We propose two architectures - Base TimeVAE and an Interpretable TimeVAE. The Base TimeVAE model is evaluated on multiple datasets and compared with current state-of-the-art timeseries data generation methods. T-SNE charts comparing similarity of generated data with original data showed that TimeVAE consistently produced high quality generated data and its performance was superior to that of other methods. Scores from discriminator and next-step prediction tests on all datasets also indicate that the TimeVAE method meets or exceeds the current state-of-the-art in time-series data generation. Finally, from Table 3 it is clear that TimeVAE requires significantly less computing time, and therefore cost, to train than existing GAN-based methods.

The TimeVAE method is also novel because it allows injection of domain-specific temporal constructs, which allows the method to produce interpretable results. Our future work on this method will focus on further proving the value of data generation methods with such temporal constructs on more complex tasks such as multi-step time-series forecasting with limited amount of data.

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
