# OpenReview forum: "TimeVAE: A Variational Auto-Encoder for Multivariate Time Series Generation"
_ICLR.cc/2022/Conference — ICLR 2022 Submitted_

### Official Review · Reviewer_f3YU · 2021-11-01

**Correctness:** 4
**Technical Novelty And Significance:** 3
**Empirical Novelty And Significance:** 3
**Recommendation:** 3
**Confidence:** 4

**Main Review:**

This paper is motivated by the limited data access for the time-series domain, so the authors propose a synthetic data generator that they claim will be valuable for deep learning research. However, I would doubt how much the proposed method can contribute to the machine learning community. In real cases, we are motived to solve problems or develop new methods based on practical challenges. It will be difficult to evaluate such a synthetical data generator can capture the necessary needs for specific questions.

1. There is no discussion about the interpretability in the results session.
2. In Table 1, there is no obvious better performance for the proposed model besides the dataset sine.

**Summary Of The Paper:**

This paper proposes an architecture for synthetically generating time-series data with the use of VAE. The authors claim that the contributions for this paper are interpretability, capable of encoding domain knowledge, and reduced training time.

**Summary Of The Review:**

The authors proposed a synthetic data generator for the time-series domain. Based on the results, I didn’t see significant performance achieved compared to the other state-of-the-art methods. Meanwhile, it didn’t discuss the interpretability of this method which is claimed as one key contribution of this method.

Strengths

Well written paper

Weaknesses

No interpretation was discussed since it was claimed as the main contribution
The overall contribution to this field is limited

---

### Official Review · Reviewer_CnbM · 2021-11-01

**Correctness:** 3
**Technical Novelty And Significance:** 2
**Empirical Novelty And Significance:** 2
**Recommendation:** 3
**Confidence:** 4

**Main Review:**

* Strengths *
The main strength of this work is that the proposed model is a sensible and general choice for learning from arbitrary time-series data. The paper is clear and generally well written, and I've found it to be technically sound. While the use of VAEs for time-series data is not novel, the proposal to combine deep architectures with interpretable components is interesting. I would suggest the authors pursue this line of investigation further as interpretability is still an open problem with deep generative models (incorporating domain knowledge can ultimately help with generalization as well). Results on the proposed datasets and proposed metrics clearly show the advantages of TimeVAE over a set of chosen GAN-based models.

* Weaknesses *
I do not think this paper is ready for publication as there are substantial limitations in its current version. One of the main issues is that this paper lacks general context of the literature in a number of areas: state-space models, deep generative models for sequential prediction tasks, particle fitlers, and so on. Its technical contribution is thus unclear: it is unlikely that there can be a single general-purpose deep generative model for time-series. Different domains require drastically different architectures and tailored learning algorithms.

- The description of the methods and the background on generative models is odd. The categorization of generative models as those that "learn from data" and those that "use Monte Carlo sampling" is confusing. I imagine the authors refer to simulation-based models (sometimes also referred to as implicit models due to the lack of a closed-form likelihood) for the second category. The explanation of VAEs is a bit nonstandard --  I would suggest framing the method as a generative model, and subsequently proposing the use of variational inference and the reparameterization trick as the way to approximately minimize the intractable marignal log likelihood.
- A number of imprecise or not very formal claims are made, such as "the computation of p(z) is very expensive and often intractable". I imagine the authors mean the posterior p(z|x)? Another example is "The GAN makes up for the deficiencies that autoregressive models have, namely that they are deterministic". Under the common definition of autoregressive models, these are probabilistic.
- To my understanding, the proposed architecture encodes the whole sequence into a global random variable modelled with a Gaussian distribution. The generative model decodes the latent vector to produce the whole output sequence. Having a single global lantent vector for a whole sequence seems like a very limiting design -- most state space model incorporate per time-step latent variable. Have the authors considered such a design?

There are also some important issues in the experimental methodology.
- While interpretable components in the decoder are introduced, these are presumably not used at all in the experiments and no evaluation is provided as to the benefit and/or intepretability of these components.
- If the purpose is to compare and find the best types of generative models for time-series, the experiments would benefit from including probabilistic autoregressive models that are generally well suited to capturing time correlations. Similarly, I'm not sure why the next-step prediction evaluation requires post-hoc learning of an LSTM model. Why not build into the VAE design the ability to predict next step (cf state-space models)?
- All the experiments are done on a number of synthetic or novel datasets unavailable elsewhere. This makes it difficult to compare the proposed method to other pieces of work in the literature.
- How does the proposed discriminator score compare to similar types of metrics that compare model samples and data samples (e.g. FID score)?

**Summary Of The Paper:**

This work proposes a generative model architecture for time-series that is trained using the variational auto-encoder framework. The main motivation for this model is (a) to obtain a more sample efficient model compared to comparable models based on generative adversarial networks (GANs), and (b) optionally incorproate domain knowledge in its design.
The authors design a fairly general-purpose VAE architecture which can also incorporate intepretable decoder blocks. Results are carried out on a number of new datasets and compared to the aformentioned GAN based time-series models. The authors use two different metrics to evaluate the mismatch between genereated and real samples, and show that their TimeVAE performs better than the alternative models.

**Summary Of The Review:**

The paper has some interesting ideas regarding incorporating domain knowledge and interpretability into a deep generative model for time-series, but falls short in a large number of aspects. The literature review is insufficient, the background and methods should be described in more precise terms, and the experiments are not applied to well-known datasets, or extensive enough such that we can robustly see the benefits of the proposed model compared to the state-of-the-art. The experiments also lack any type of evaluation of the interpretability aspect of the proposed model, thus I do not see it as an important contribution. Overall, I suggest a major revision to the paper before recommending its acceptance.

---

### Official Review · Reviewer_X2k3 · 2021-11-02

**Correctness:** 3
**Technical Novelty And Significance:** 1
**Empirical Novelty And Significance:** 2
**Recommendation:** 3
**Confidence:** 4

**Main Review:**

The paper is well written and I think clearly conveys the main points of their models as well as the related work.

Primarily however, it is not clear to me why the proposed models (and definitely the proposed "base" model) are not just essentially a standard VAE (or conditional VAE) applied to time series data. The architecture even includes convolutional layers (which yes have been showed to be sometimes useful for time series task) as opposed to maybe including some form of recurrent layer which might distinguish it more - this seems a bit strange.

A counter might be that "well TimeGAN is just a GAN applied to time series data" but there at least they introduce additional loss functions in order to try and capture stepwise marginal distributions as well for example, there doesn't appear to be any of that sort of thing in this paper which makes it hard to see how it can be claimed that TimeVAE better captures the temporal aspects of the data.

The VAE loss function is just presented as is with little discussion as to exactly how it comes about. I'm sure most readers will be familiar, but I think it's worth discussing it's derivation. This is especially the case as the paper goes on to actually weight the terms, and a further discussion as to why that's appropriate or reasonable would be useful.

The empirical evaluation I think is done well and is compelling that TimeVAE does produce realistic time series trajectories - they could use an expanded set of benchmarks though given how many methods there are out there in the generative models space - the TimeGAN for example contains a number of extras that are not considered here.

Equations should be formatted correctly in the middle of the line and numbered.

**Summary Of The Paper:**

The paper presents a method for a generative model of time series data based on variational auto-encoders. They suggest two models, a simplified base version as well as a conditional "interpretable" one that can be used to customise trend and seasonality aspects of the generated data.

**Summary Of The Review:**

The proposed methods simply do not seem to be presenting any sort of substantial originality that would make for me to recommend acceptance.

---

### Official Review · Reviewer_Gi2q · 2021-11-02

**Correctness:** 3
**Technical Novelty And Significance:** 3
**Empirical Novelty And Significance:** 2
**Recommendation:** 3
**Confidence:** 4

**Main Review:**

### Strengths
- Well written paper
- Interesting approach to include methods from “classical” timeseries

### Weaknesses
- The introduced decoder building blocks seem to also introduce a set of new hyper-parameters. There are no details provided on how to best select these hyper-parameters nor how to perform model selection regarding these hyper-parameters.
- The authors motivate the newly introduced building blocks with interpretability. In the empirical evaluation, there is no experiment or ablation that would provide any evidence on how interpretable the newly created building blocks are.

### Questions
- What about GP-VAE (Fortuin et al: Deep Probabilistic Multivariate Time Series Imputation, 2020)? How related is time-series imputation to next step prediction? I would be interested in the authors thoughts on that.
- Table 1: I am not sure if I read the results correctly in Table 1. According to the description on the discriminator score in section 4.3. a value close to zero indicates a good performance of the model (the score is 0.5 - accuracy). For the 100% version of the training set, performance numbers are often worse, than for the 2% version of the training set. This is counter-intuitive to me. Could you please elaborate on that?
- Are the t-SNE plots bin figure 4 generated using reconstructions of samples or randomly generated samples? I could not find the respective information.

**Summary Of The Paper:**

The authors present a VAE for multivariate time series generation. The authors add additional decoder building blocks modelling level, trend and seasonality components of time series. Their method, named TimeVAE, is evaluated on four different datasets. Performance measures of generated samples are the similarity to real samples and the predictive quality of generated, “future” time steps.

**Summary Of The Review:**

The authors provide a simple and interesting extension to VAEs for multivariate timeseries. Although I like the idea of the paper, the limited empirical evaluation can be improved in my opinion. Although the authors perform experiments on 4 datasets to compare to previous works, the hyper-parameters of the core building blocks are not evaluated properly. Ablation studies and the selection of values are missing. Therefore, this paper is not ready for publication at ICLR in my opinion.

---

### Decision · Program_Chairs · 2022-01-20

**Decision:**

Reject

**Comment:**

The authors propose a VAE-based architecture for generating multivariate time series. The base version of TimeVAE models a distribution over a fixed-length sequences of observations using a latent vector of fixed dimensionality and a convolutional encoder and decoder. The Interpretable TimeVAE model incorporates additional features from traditional time series models such as explicit modelling of trends and seasonality. TimeVAE is compared to several baselines such as TimeGAN on four small times series dataset and seems to perform competitively according to two custom evaluation metrics and a visualization.

The reviewers thought that the paper was interesting but not ready for publication due to the following:
-The paper's contributions and their significance are not clear
-Interpretable VAE was not used in the experiments and its interpretability has not been verified
-Coverage of related work is insufficient